# Anisotropic Optical Response of WTe_2_ Single Crystals Studied by Ellipsometric Analysis

**DOI:** 10.3390/nano11092262

**Published:** 2021-08-31

**Authors:** Krastyo Buchkov, Rosen Todorov, Penka Terziyska, Marin Gospodinov, Velichka Strijkova, Dimitre Dimitrov, Vera Marinova

**Affiliations:** 1Institute of Optical Materials and Technologies, Bulgarian Academy of Sciences, 1113 Sofia, Bulgaria; k.buchkov@hotmail.com (K.B.); rntodorov@abv.bg (R.T.); vily@iomt.bas.bg (V.S.); 2Institute of Solid State Physics, Bulgarian Academy of Sciences, 1784 Sofia, Bulgaria; penka.terziyska@gmail.com (P.T.); maringospodinov@abv.bg (M.G.)

**Keywords:** WTe_2_, chemical vapor transport, XRD, Raman spectroscopy, spectroscopic ellipsometry

## Abstract

In this paper we report the crystal growth conditions and optical anisotropy properties of Tungsten ditelluride (WTe_2_) single crystals. The chemical vapor transport (CVT) method was used for the synthesis of large WTe_2_ crystals with high crystallinity and surface quality. These were structurally and morphologically characterized by means of X-ray diffraction, optical profilometry and Raman spectroscopy. Through spectroscopic ellipsometry analysis, based on the Tauc–Lorentz model, we identified a high refractive index value (~4) and distinct tri-axial anisotropic behavior of the optical constants, which opens prospects for surface plasmon activity, revealed by the dielectric function. The anisotropic physical nature of WTe_2_ shows practical potential for low-loss light modulation at the 2D nanoscale level.

## 1. Introduction

Tungsten ditelluride (WTe_2_) is one of the most prominent layered transition metal dichalcogenide (TMD) materials, with diverse and complex physical properties, providing many unexplored research directions in both fundamental and applied materials science [1,2,3]. These are related to its unique type-II Weyl fermion nature; it is a topological semimetal in which both edge and 2D bulk metallic states exist, with the potential to be modified in a 2D topological insulator using various control options such as temperature, magnetic, electric and optical fields, pressure and strain, electrostatic gating and bias voltages [4,5,6,7,8].

The origin of the electronic band nature of WTe_2_ [9] is a distinct orthorhombic 1Tʹ (distorted) crystal structure, also known as the T_d_ phase—in contrast to the usually observed 2H and 1T variants in TMD-class materials. The W atoms are arranged in a distorted triangular lattice (between the consecutive 180°-rotated Te layers) with the formation of quasi(1D) W-W chains, leading to significant tri-axial anisotropy [10,11] of the electrical, magnetic, thermal and optical properties.

The complexity of WTe_2_ physics also includes unique phenomena such as titanic (non-saturating) magnetoresistance [12], “negative” magnetoresistivity [13], a room temperature ferroelectric semimetal state [14], superconductivity [15] and a diverse set of electronic and quantum topological effects in 2D form, such as the temperature-induced Lifshitz transition [16], the linear Nernst effect [17] and the observation of the quantum spin Hall effect up to 100 K [18].

In the photonics field, 2D layered materials [19,20], and particularly the TMD family [21], offer many options for functional light control, strong light–matter interaction effects and reduced optical losses. We have to emphasize their notable optical anisotropy [22,23], which is a key factor in light manipulation due to the bi/trirefringence and di/trichroism effects [24], which result in spatial and polarization separation, enabling a virtually unlimited platform for the realization of the next generation (4G) of planar optics.

For the TMD materials group, the large in-plane vs. out-of-plane anisotropy arises from different interlayer (weak van der Waals bonds) and intralayer strong atomic interactions due to ionic/covalent bonding. Consequently, record birefringence values were observed in MoS_2_ [22], as high as Δn~1.5 in the infrared and Δn~3 in the visible spectral range. Currently, such high birefringence is offered by artificially designed metamaterials and metasurfaces; however, the fabrication processes and optical losses are the main challenges [25]. For example, inorganic solids and liquid crystals possess relatively small birefringence, typically up to 0.4 [26,27]. At the same time, several 2D materials such as “black phosphorous” [28] and the monochalcogenides family [29] display exceptional tri-axial optical anisotropy (theoretically predicted for ReSe_2_ [30] (isostructural to WTe_2_)).

In this context, WTe_2_ is intensively studied with comprehensive spectroscopic investigations of the electronic band structure. The uniaxial shaped Fermi surface and inter-band transitions lead to significant intrinsic anisotropy [10,11,31,32] and, together with the 2D layered topological [33,34,35,36] properties, expand the list of degrees of freedom for the optical control and detection of the light wave-vector, the intensity, polarization, phase, frequency, nanoscale confinement and collective charge excitations (plasmonic activity) [37,38,39,40,41]. Optical absorber devices based on WTe_2_ have already been developed as part of prototype ultrafast laser systems [42,43,44].

Moreover, the synthesis of WTe_2_ is rather difficult due to low chemical reactivity between W and Te compounds. The production of ultrathin WTe_2_ nanostructures via conventional thin-film deposition techniques such as chemical vapor deposition [45] and molecular beam epitaxy [46] is quite limited and currently realized in a mono- or few-layered (2D) form only at µm-scale clusters. Alternatively, the unique properties of WTe_2_ at the nanoscale level are mainly accessible via exfoliation techniques, which makes the development of the bulk crystal growth process of prime importance.

Following the above-noted research trends and perspectives, here we present an unexplored view of the tri-axial anisotropic optical response of WTe_2_ (cleaved high-quality single crystals) via a study of reflection ellipsometry. The paper is structured as follows: we first report extended details about the growth process of high-quality WTe_2_ single crystals using the CVT technique. Next, we briefly present the WTe_2_ structure confirmed by X-ray diffraction (XRD), optical profilometry and Raman analysis. As a final point, we discuss the behavior of the main optical constants and dielectric function components, depending on the crystallographic orientations.

## 2. Materials, Methods and Experimental Details

### 2.1. WTe_2_ Crystal Growth Procedure Using the CVT Method

WTe_2_ single crystals were grown using the chemical vapor transport (CVT) method, a conventional approach for the preparation of TMD materials [47,48]. The preliminary stages start with the mixing of W and Te precursors (high purity > 5 N) for a solid-state reaction synthesis at 725 °C (72 h) in an evacuated ampule (~10^−4^ torr), followed by additional powdering and homogenization steps. The pre-sintered WTe_2_ material and an appropriate amount of Br (a volatile transport agent to mediate the growth process—3 mg/mL volume) were sealed in evacuated CVT quartz ampule. The ampule was designed with enlarged diameter, considering the geometry of the dual-zone furnace thermal gradient, the W/Te phase diagram [49], the precursor amounts required to maintain the optimal thermal equilibrium and the crystallization conditions for the formation of large-sized WTe_2_ crystals. Finally, the ampule was positioned in separate thermal gradient zones of 780 °C and 680 °C and the growth process was maintained for a maximally extended period—approximately 190 h. The CVT process is schematically visualized in Figure 1.

The obtained WTe_2_ crystal flakes had an elongated rectangular geometry as a result of the high a/b orthorhombicity ratio. Their size varied from mm to cm, with a flat and shiny metallic surface, and their thickness was usually under 100 µm. The residual traces of Br transport contaminations were removed via dynamic evacuation for 24 h, followed by a simple surface cleaning procedure of immersing the crystals subsequently in acetone (5 min) and in isopropanol (5 min).

WTe_2_ is vulnerable to ambient conditions, with rapid and irreversible surface oxidation [14,50,51]. Therefore, the samples were stored in GloveBox system (under inert atmosphere with oxygen/moisture levels less than 1 ppm) and exposed only during the experimental investigations. In order to provide a more unperturbed view of the intrinsic optical properties of the material, the crystal samples were cleaved with a Nitto tape to attain a fresh surface just before the ellipsometric experiments.

### 2.2. Characterization Instruments and Techniques

The crystal structure of powdered WTe_2_ crystals were analyzed using XRD over the 2θ range of 5.3° to 80°/constant step of 0.02°, using a Cu K_α_ “D8 Advance” Bruker (Karlsruhe, Germany) diffractometer equipped with a LynxEye detector. The structural data were identified with Diffracplus EVA using the ICDD-PDF2 crystallographic database. The WTe_2_ crystal structure was presented using Vesta ver. 3.4.8 [52] structural analysis and visualization software.

The surface morphology was investigated in several sectors using an optical profilometry/metrology system, Zeta-20, from Zeta Instruments. The Raman spectra were recorded (backscattering geometry) using a HORIBA Jobin Yvon Labram HR visible spectrometer with a He-Ne laser (633-nm excitation line) and a Peltier-cooled charge-coupled device (CCD) detector.

The ellipsometry measurements were performed in reflection mode using a J.A Woollam (Lincoln, NE, USA) M2000D rotating-compensator spectroscopic ellipsometer in the spectral range from 193 nm to 1000 nm.

Complete EASE software (J.A Woollam) was used to measure and model the spectra of the ellipsometric Ψ and Δ parameters. The analysis procedure is similar to that described in the case of a biaxial microcrystal of MoS_2_ [22]. As a last step in fitting the measured spectra of ψ (the amplitude ratio) and Δ (phase difference), we determine the Euler angles—the rotation of the crystal axes (n_x_, n_y_, n_z_) relative to the experiment coordinate system, denoted by *N*, *S* and *P* (where *N* is the normal to the surface of the sample and *S* and *P* are the directions perpendicular and in the plane of incidence of light). The values obtained for the Euler angles were less than 1.8°.

For the specific investigation and analysis (XRD, Raman spectroscopy and Ellipsometry), several high-quality WTe_2_ crystal flakes were selected (using optical microscopy and profilometry, Appendix A) depending on their defect-less surface quality. Due to their fragile and delicate morphology (resembling an ultrathin foil), the samples were carefully attached to a glass substrate to ensure the effective implementation of ellipsometric experiments.

## 3. Discussion and Results

### 3.1. XRD Analysis

The crystal structure of the selected samples was characterized using X-ray diffraction. Powder XRD analysis was used and the results are presented in Figure 2. The phase identification did not detect crystalline impurities. The cell parameters were determined to be: a = 6.278(6) Å, b = 3.483(5) Å and c = 14.054(8) Å, which are typical for the designated orthorhombic space group: Pmn21, No. 31 [1,53]. The main diffraction peak [002] was detected at 12.60° 2θ degrees, whereas, due to its layered structure, the other characteristic peaks [004], [006], [008] and [0010] were detected with minor intensity. The preferential [00l] crystallographic orientation was a fine confirmation of the highly crystalline nature of the samples and the efficiency of CVT growth.

### 3.2. Raman Spectroscopy

In addition to the XRD, the single crystal structural quality was also indirectly assessed by means of Raman spectroscopy analysis. Raman peaks (presented in Figure 3.) typical for pure WTe_2_ are identified as $A_{1}^{9}$~210 cm^−1^, $A_{1}^{7}$~163 cm^−1^, $A_{1}^{4}$~132 cm^−1^, $A_{1}^{3}$~111 cm^−1^ and $A_{2}$~90 cm^−1^ [32,54]. Usually, narrow and intense Raman peak structures are related to the ability of the long-range propagation of Raman phonon modes through the crystal lattice, marking a high crystallinity level.

In addition, the crystal surface morphology and topology was investigated by means of optical profilometry (Figure A1) which verified the quality of the crystals selected for further cleaving and measurements.

### 3.3. Spectroscopic Ellipsometry

Optical anisotropy is a key factor in light manipulation due to the bi/trirefringence and di/trichroism effects, which results in spatial and polarization separation, enabling a new degree of freedom for next-generation photonic devices. Usually, for the anisotropy analysis of 2D TMD materials, the spectroscopic ellipsometry is applied in a transmission regime, considering the Jones matrix formalism and the finite (non-zero) values of the off-diagonal components [55,56] or using ellipsometric imaging [22] for thicker samples ~1 µm due to the extreme signal dominance of the in-plane vs. out-of-plane optical components. However, due to the high values of the extinction coefficient k(*λ*) of WTe_2_ [38] (resulting in a small light penetration depth) and the narrow band gap, high optical absorption is expected for the wavelengths shorter than 2000 nm, which makes the use of transmission ellipsometry difficult in the visible and ultraviolet spectral regions. In addition, considering the noted limitations of WTe_2_ synthesis in large-scale thin films, we applied reflection ellipsometry for the qualitative comparison and analysis of WTe_2_ crystal flakes.

For the modeling of ellipsometric data, we used two different models. The initial analysis model assumed that the sample is an isotropic medium and two independent measurements were made at each incident light angle, considering the crystallographic orientation along the *a-* and *b*-axes through the elongated sample geometry, with the second measurement being performed by rotating the crystal to 90°. The dispersive refractive index $n\left(\lambda \right)$ and extinction coefficient $k\left(\lambda \right)$ were extracted from the direct inversion of the experimental ellipsometric ψ (the amplitude ratio) and Δ (phase difference) functions, associated with the p- and s-polarized light waves’ characteristics.

The second approach was based (using the same experimental data set) on the assumption that the sample had an anisotropic triaxial crystal symmetry with different refractive indices along the three mutually perpendicular axes. Concerning the dielectric permittivity, we applied the Tauc–Lorentz model [57].

#### 3.3.1. Basic Isotropic Model Approach: Refractive Index and Extinction Coefficient

For the applied isotropic model, the spectral behavior of the fundamental optical constants of WTe_2_ crystals are presented at Figure 4 and Figure 5. The refractive index $n\left(\lambda \right)$ for the cleaved WTe_2_ crystal reached typically high values (~4) in both orientations (parallel to the *a* and *b* directions) with a specific double slope increasing with the wavelength (more rapidly in the ultraviolet region). Based on the observed n(λ) values, WTe_2_ can be categorized as a crystal with high-index dielectric class material (compared to other layered chalcogenides [58,59,60]), with a corresponding prospect for light absorption enhancement and directional light manipulation in photonic device applications.

The extinction coefficient k(λ) also changed noticeably in both in-plane crystallographic directions (Figure 5). Different trends of the optical losses were observed in the investigated spectral area. Along the *a*-axis, the values of k(λ) varied in the range of 1.5–3 with an observed minimum at ~650 nm, whereas, for the *b*-axis, there was a gradual decrease from 3 to 1.

The performed optical measurements showed significant incident angle dependencies, resulting from different signal contributions along the orthogonal axes, mostly in the near-infrared region. Generally, this is also related to the typical anisotropic behavior of light propagation in low-symmetry layered materials [61].

Due to the ambient environment vulnerability of WTe_2_, which leads to the formation of a thin oxidation layer at the native surface, the possible effect of this on the optical constants was also studied. The optical response of the native WTe_2_ crystal surface (before the cleaving) is presented in Figure A2A–D.

#### 3.3.2. Tauc–Lorentz Oscillator Model: Refractive Index and Extinction Coefficient

The notable tri-axial anisotropic behavior of WTe_2_ was evident through numerical analysis using a Tauc–Lorentz oscillator model [56,57]. This method is generally applied for comprehensive ellipsometric investigations of the optical constants for layered TMD materials [22,61] and it is based on combined dielectric function parametrization using the Tauc Law and the Lorentzian oscillator model. For the particular tri-axial case of WTe_2_, the dielectric tensor elements were described by the Tauc–Lorentz formalism considering different model parameters (fitting variables: $E_{g}$ (optical band gap), Lorentz oscillator amplitude and energy, broadening factor) in the crystallographic directions. The number and the initial values of the frequencies of the Lorentz oscillators are determined by the local maxima observed in the spectra of the imaginary part of the complex dielectric constant obtained by the isotropic model. The results of the analysis showed that the minimum number of Lorentz oscillators required to obtain the best fit by the anisotropic model was three in the directions of the *a*- (at 1.07 eV, 2.98 eV and 5.01) and *b*-axes (at 2.87 eV, 3.84 eV and 4.77 eV) and two in the direction of the *c*-axis (at 4.65 eV and 5.84 eV).

The calculated dispersions, with significant qualitative and quantitative differences in the refractive index and extinction coefficient along the three (mutually perpendicular) axes, are presented in Figure 6 and Figure 7. In addition, the relative (Figure 6, inset graph) in-plane vs. out-of-plane anisotropy ratio $∆n=\bar{n_{ab}}-n_{c}$ and birefringence was determined, reaching a maximal value ~1.5 in the studied visible and near-infrared region, along with the averaged in-plane $\bar{n_{ab}}$ response. For clarity, the data are presented in the visible/near-infrared spectral range due to the presence of resonant peaks in the ultraviolet region.

The extrapolated values of the optical band gap $E_{g}$ in the *a* and *b* directions were 0.006 eV and 0.273 eV, respectively. The obtained value of 0.006 eV for the band gap along the *a*-axis suggests a metallic character of the crystal in this direction and high values of the extinction coefficient, whereas the larger values of $E_{g}$ in the other direction suggest a decrease of $k\left(\lambda \right)$ values which become zero at a photon energy corresponding to the relevant band gap width. These results are in good agreement with the values determined by the density functional theory and first-principle calculations of the band structure reported in the literature [1,62].

The observed trirefringence and trichroism scenarios, with high unidirectional control of the light propagation, together with the high $n\left(\lambda \right)$ and low $k\left(\lambda \right)$ values (expected to be even higher for the exfoliated and thinner layers) are very promising for the development of emerging nanophotonic devices. The Tauc–Lorentz model also showed significant optical constant differences calculated at different angles of light incidence, especially at shorter wavelengths, as shown in Figure 7 (inset), for the extinction coefficient along the *b*-axis.

We assume that this notable angle dependence difference is also related to the narrow light penetration depth expected for WTe_2_. Considering the Beer–Lambert–Bouguer law: $T\sim e^{-\alpha d}$ (where T is transmittance, *α* is the absorption coefficient linked with the extinction coefficient *α* = 4πk/λ and d is the the thickness of the sample), we can estimate the expected penetration depth for WTe_2_ (presented in Figure A3, defined as αd ≤ 1).

The spectra of the extinction coefficient $k\left(\lambda \right)$ show that the penetration depth decreases with decreasing wavelength. The contribution from the *c*-axis decreases and, in this case, the simplistic isotropic model gives a better match in determining the refractive index and the extinction coefficient at the different incidence angles.

At longer wavelengths, the extinction coefficient decreases and consequently the penetration depth of the electromagnetic wave increases. Considering the axial influence, these will give a better match when counted at different angles. Naturally, the isotropic model does not take into account that the refractive index along the *c*-axis is different. Therefore, this leads to a difference in the optical constants (Figure 4 and Figure 5) at wavelengths longer than 650 nm.

#### 3.3.3. Dielectric Functions

In addition, based on the Tauc–Lorentz oscillator model and anisotropic analysis of the fundamental optical constants, the values of the real $e\prime \left(\lambda \right)$ and imaginary $e\prime \prime \left(\lambda \right)$ dielectric function components were determined for the three crystallographic directions. For the cleaved WTe_2_ crystal, the anisotropic trends are presented in Figure 8.

The main feature is the observation of negative values for $e\prime \left(\lambda \right)$ [37] in the ultraviolet/visible edge, revealing the presence of plasmonic resonances mediated by strong inter-band electronic transitions (as an alternative mechanism to the conventional collective excitation of the free charge carriers), typically observed in semi-metals (such as WTe_2_), semiconductors and topological insulator materials [63].

Consequently, the sophisticated (low-symmetry Fermi surface) electronic band structure of Weyl (and Dirac) semi-metal materials also results in the observed notable anisotropy of the optical properties. The corresponding highly directional modulations of the carrier density and mobility also leads to the significant differences in the light polarizability and the dielectric functions along the *a*-, *b*- and *c*-axes in the case of WTe_2_ single crystals.

## 4. Conclusions

In summary, large-sized WTe_2_ crystals were grown by using the CVT method. Their crystal structure and morphology were characterized using XRD, optical profilometry and Raman spectroscopy, verifying their high crystallinity and surface quality. The optical response of cleaved WTe_2_ crystal samples was analyzed by means of reflection spectroscopic ellipsometry. Two different data models were implemented—a simplistic isotropic and a Tauc–Lorentz based anisotropic model, revealing:
High values of the refractive index (~4) were observed for both models, especially in the visible/near-infrared spectral range. The influence of the optical anisotropy was also well detected with the simplistic isotropic modeling, with observed distinct differences in the optical constants (n and k) depending on the in-plane crystallographic orientations.Identical anisotropic tendencies of the intrinsic WTe_2_ spectral behavior were also observed for the native crystal surface, which is expected to be affected by the formation of an oxide surface layer (Appendix A).The calculated dispersions of the refractive index and extinction coefficient in the frames of the Tauc–Lorentz oscillator model indicate the tri-axial nature of the optical response.The extrapolated band gap values of 0.006 and 0.273 eV in the *a* and *b* directions, respectively, are in good agreement with those determined by energy band structure calculations in the literature.The real component of the dielectric function shows negative values, which also indicates surface plasmon polariton activity.

The observed anisotropic optical response emphasizes the practical prospects of WTe_2_ for polarization-dependent light modulation and dielectric plasmonics in the innovative field of nanophotonics.

## Figures and Tables

**Figure 1 nanomaterials-11-02262-f001:**
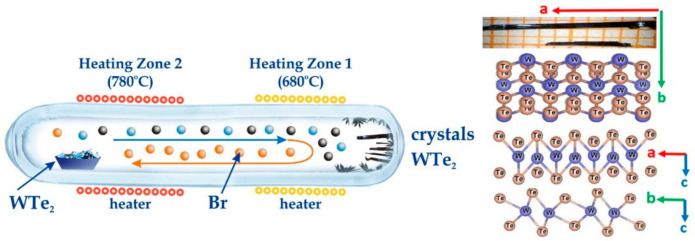
Schematic of the CVT crystal growth process (left). Optical images of the largest acquired samples (top right) with the crystallographic axis orientations and a visual representation of the layered crystal structure of WTe_2_ (using Vesta).

**Figure 2 nanomaterials-11-02262-f002:**
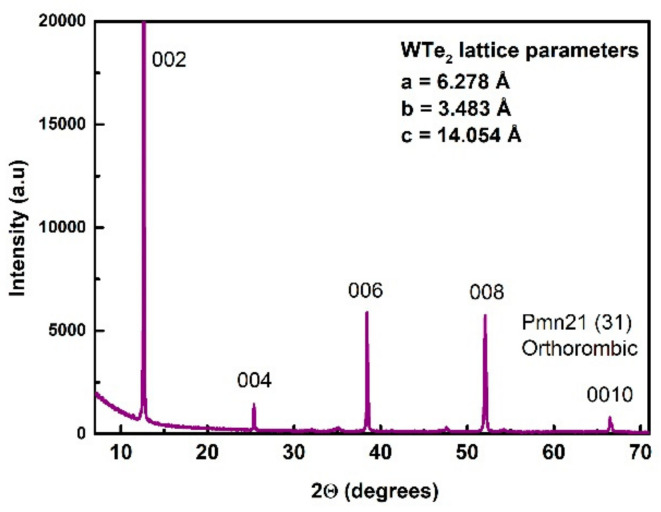
XRD analysis of WTe_2_ sample with the main identified peaks of the preferential crystallographic orientation.

**Figure 3 nanomaterials-11-02262-f003:**
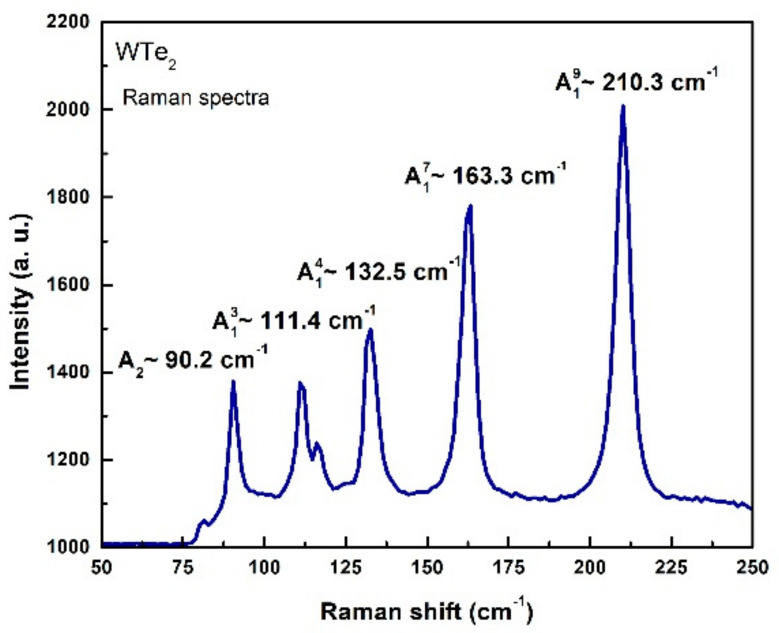
Raman spectrum of WTe_2_ single crystal.

**Figure 4 nanomaterials-11-02262-f004:**
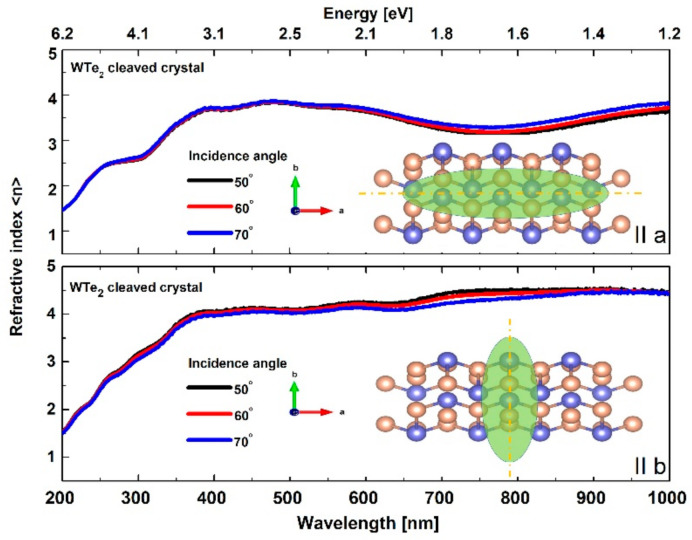
Spectral dependence of refractive index in the a and b directions for the cleaved WTe_2_ crystal.

**Figure 5 nanomaterials-11-02262-f005:**
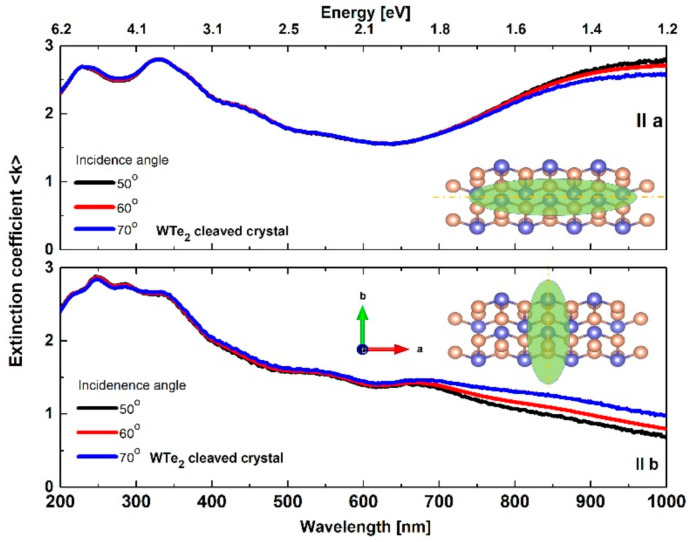
Spectral dependence of extinction coefficient in the a and b directions for the cleaved WTe_2_ crystal.

**Figure 6 nanomaterials-11-02262-f006:**
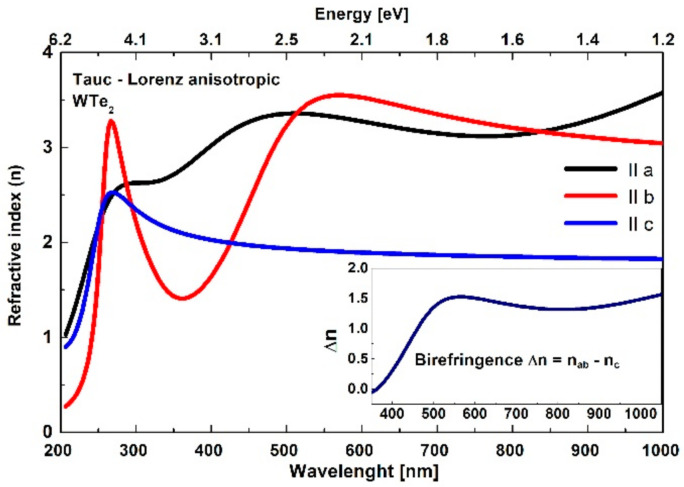
The tri-axial optical anisotropy of the WTe_2_ refractive index in the frames of the Tauc–Lorentz model. Inset: relative birefringence in the visible/near-infrared region.

**Figure 7 nanomaterials-11-02262-f007:**
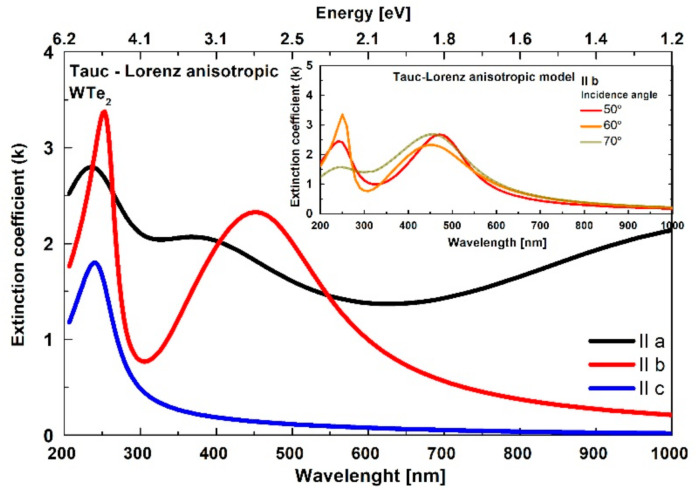
The triaxial optical anisotropy of the WTe_2_ extinction coefficient in the frames of the Tauc–Lorentz model. Inset: incidence angle dependence of the extinction coefficient along the *b*-axis.

**Figure 8 nanomaterials-11-02262-f008:**
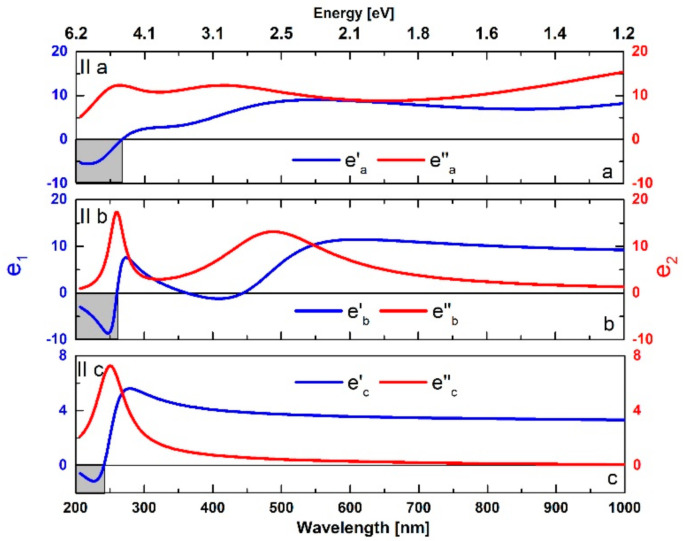
The tri-axial spectral behavior of the dielectric functions for WTe_2_.

## Data Availability

The data presented in this study are available on request from the corresponding and leading authors.

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
