# Peer review of "Anisotropic Optical Response of WTe2 Single Crystals Studied by Ellipsometric Analysis"

_nanomaterials, 2021, doi:10.3390/nano11092262_

Round 1

Reviewer 1 Report

The authors propose a WTe2 grown by CVT technique. They analyzed the WTe2 crystal by XDR, Raman spectroscopy, and spectroscopic ellipsometry. The anisotropy was confirmed by Tauc-Lorenz model. The results showed the synthesized WTe2 has a high refractive index. The anisotropy and dispersion are studied. The material is useful for next-generation photonic.

The experiment flow and results are well organized. The discussions are clear and explicit. However, I suggest the author discuss more information about the state of the art for the current synthesis of WTe2 to emphasize their advantage.

Author Response

Dear MDPI Nanomaterials Editors,

We would like to present a revised version of the submitted manuscript (Buchkov et al. Nanomaterials 1283349) entitled "Anisotropic Optical Response of WTe2 Single Crystals Studied by Ellipsometric Analysis.

We are very grateful to the referees for the critical reading and
valuable recommendations for further improvement of the manuscript. We would like to present the list of replies to the referees’ comments.

We have made our amendments in the manuscript respecting the Referees comments.

REPLY TO THE REFEREE REPORT:

Referee 1

Referee 1 Suggestion. “However, I suggest the author discuss more information about the state of the art for the current synthesis of WTe2 to emphasize their advantage.”

We are grateful to the referee for this suggestion, since we have started with extensive preliminary test/trial process for acquiring the optimal WTe2 crystal growth conditions: using large (as possible) quartz ampule diameter and the (custom made) furnace manufactured especially for the WTe2 synthesis requirements. In addition, the growth period duration is also extended.

Following the remarks, we modified the Crystal Growth related paragraphs by strengthening these important points.

Reviewer 2 Report

My comments are in attached file.

Author Response

Referee  2

Referee 2 Suggestions and Correction Remarks

We are sincerely grateful to the Reviewer for providing a very detailed list with suggested corrections for grammar and text readability improvement. The corresponding text alterations are included in the revised manuscript version.

Several questions are pointed out by the Referee

  • How is it possible to record XRD pattern from single crystal?

Usually, the Laue diffraction method is applied for structural analysis of single crystal samples. However, we used the more conventional (and available) but destructive method “Powder” XRD, which required the test samples to be in powdered form.

How is it possible to check chemical composition by Raman spectroscopy?

We have cleared the context of paragraphs according to the reviewer recommendation. Our considerations were based on the high sensitivity of the Raman spectrum to the crystal structure that can be affected by eventual stoichiometry deviations or the noted oxidation effects (as pointed out in Scientific Reports 5 10013 (2015) and Fig. 5b).

  • For generality, the dispersive refractive index obtained in WTe2 could be compared to the results reported for other layered chalcogenide crystals:

  • Growth Des. 11 (2011) 5507-5514
  • Chem. Phys. 146 (2014) 12-17
  • Solid State Chem. 236 (2016) 203-208

Following the recommendation, we have updated the reference list including the comparison studies (references) of the dispersive refractive index for other topological materials.

Reviewer 3 Report

The manuscript entitled Anisotropic Optical Response of WTe2 Single Crystals Studied by Ellipsometric Analysis written by K. Buchkov et al. reports on fabrication and ellipsometric characterization of optically anisotropic WTe2 single crystals.

The manuscript contains sound and comprehensive Introduction where the need for determination of WTe2 optical constants is clearly evidenced. On the other hand, Section 3. Discussion and Results should be substantially improved.

Consider for example following points:

  1. Optical characterization of anisotropic crystals is relatively demanding task. In the case of tri-axial WTe2, determination of three complex refractive indices for each wavelength is performed . This requires adequate number of independent experimental ellipsometric data and their proper treatment. The performed experimental procedure and data treatment should be presented in more details in the manuscript to prove validity of the obtained optical constants.
  2. I do not clearly understand the meaning and application of Tauc-Lorentz oscillator model that is mentioned in Section 3.3.2. Tauc-Lorentz parameterization of dielectric function was designed for amorphous materials, not for crystalline solids. Here, one should rather use combination of Drude and Lorentz oscillators in a similar manner as it is done for example in the reference [31]. Moreover, the parameterization of dielectric function that has been used should be presented explicitly in the manuscript (including information about the number of oscillators and their type).
  3. What is the benefit or reason for the approximate Basic isotropic model approach presented in Section 3.3.1. if all three diagonal components of dielectric tensor (entirely describing optical anisotropy of WTe2 ) have been determined in Section 3.3.2?
  4. To perform a consistence check of presented optical constants in Sections 3.3.1 and 3.3.2., it would be convenient to compare (for each incidence angle) the pseudo-dielectric constants determined in the frame of the Basic isotropic model approach with those calculated from complex refracted indices presented in Figures 6. and 7.
  5. In the line 248 you write

The main feature, the observation of negative values for ?′(?) [37] in the ultraviolet/visible edge reveal the surface plasmon polariton activity due to the resonance coupling between the phase of the incident light and the collective excitations of the free carriers.

but spectral features in UV are due to inter-band electronic transitions (realized by bound electrons), therefore, please, explane in more details what do you mean by … the collective excitation of the free carriers...

  1. Meaning of the sentence from the line 252

Consequently, the strong optical anisotropy of the dielectric response has a distinct difference in case of single crystals: for a-, b- and c- axis as expected for a Weyl (and Dirac) material with such high uniaxial nature of the Fermi surface and the corresponding directional dielectric polarizability of the carrier inter-band transitions and concentration.

is not clear for me.

  1. Meaning of the sentence from the line 160

…the spectroscopic ellipsometry is applied in a transmission regime due to the requirements of Fresnel matrix formalism [55] …

is not clear for me.

Author Response

Referee 3 Remark 1

Optical characterization of anisotropic crystals is relatively demanding task. In the case of tri-axial WTe2, determination of three complex refractive indices for each wavelength is performed. This requires adequate number of independent experimental ellipsometric data and their proper treatment. The performed experimental procedure and data treatment should be presented in more details in the manuscript to prove validity of the obtained optical constants.

Authors response

We are grateful for this remarks and would like to point out that one of the advantages of the studied WTe2 sample is that the orientation of the main crystallographic axes is known in advance and the high absorption coefficient does not require the removal of the reflection from the back of the sample.

Following the Reviewer remark, we have included extended description of the Ellipsometric data analysis in the Experimental section paragraph

Referee 3 Remark 2

I do not clearly understand the meaning and application of Tauc-Lorentz oscillator model that is mentioned in Section 3.3.2. Tauc-Lorentz parameterization of dielectric function was designed for amorphous materials, not for crystalline solids. Here, one should rather use combination of Drude and Lorentz oscillators in a similar manner as it is done for example in the reference [31].

Moreover, the parameterization of dielectric function that has been used should be presented explicitly in the manuscript (including information about the number of oscillators and their type).

Authors response and discussion

For the particular choice for appropriate dispersion models (Drude, Lorentz, Drude-Lorentz and Tauc-Lorentz) we have considered that the Drude model describes well the metal type dispersion in the infrared spectral region characterized by the presence of free electrons, while the Lorentz model is used to describe inter-band transitions (valence-band gap transitions). Nevertheless, as pointed out by the Reviewer (in the following remarks) there is no detected contribution of the free electrons in WTe2.

Therefore, we have decided to use the advantageous combination of the Tauc’s rule with Lorentz model considering the absorption edge of investigated samples. Tauc's rule is used to describe the dispersion of the absorption coefficient regardless of the type of material (crystalline or amorphous), when using it is more important to know the type of transitions - direct or indirect. In the Tauc-Lorentz model, Eg as band gap parameter only defines the photon energy for which the absorption coefficient becomes 0. In existing publications, it is indicated as suitable not only for amorphous materials. Another advantage of the Tauc-Lorentz model is that there are two connections between the variance of the complex dielectric function and the band gap. The lack of data on the complex dielectric function or refractive index is the main reason to compare other results, such as the band gap with existing data in the literature.

From this point of view, the Tauc-Lorentz model was successfully applied to describe dispersion of complex dielectric function of the ellipsometric analysis of metal di-chalcogenides (Ermolaev et al. Nat. Commun. 12, 854 (2021), G.-H. Jung et al. Nanophotonics 2019; 8(2): 263–270) and other crystalline materials (M. Zhao et al., Optical Materials 102 (2020) 109807).

We have modified the text with more explicit description of the parameterization of the dielectric function which is also considered in the following discussions.

Referee 3 Remark 3

What is the benefit or reason for the approximate Basic isotropic model approach presented in Section 3.3.1. if all three diagonal components of dielectric tensor (entirely describing optical anisotropy of WTe2 ) have been determined in Section 3.3.2?

The measurements in two mutually perpendicular directions is a standardized procedure for studying anisotropic optical medium. It gives information whether two of the axes lie in the plane of the sample. As we noted in the previous discussion, the advantage of the studied sample is the knowledge of the direction of the crystal axes (two of which are perpendicular to the plane of incidence).

We agree with the reviewer's note that the study of anisotropic samples is a difficult task and reasonably lead to our choice to examine a sample with a certain orientation of the crystallographic axes. In addition, the isotropic model provides information on the effective refractive index and allow also to determine the initial values of the frequencies of the Lorentz oscillators and their number (noted also in Remark 2).

Following the Reviewer remark, we have modified the Ellipsometric analysis text in order to include the dielectric function parametrization data (photon energies of the Lorenz oscillators determined in the frames of the isotropic model). 

Referee 3 Remark 4

To perform a consistence check of presented optical constants in Sections 3.3.1 and 3.3.2., it would be convenient to compare (for each incidence angle) the pseudo-dielectric constants determined in the frame of the Basic isotropic model approach with those calculated from complex refracted indices presented in Figures 6. and 7.

We are very grateful to the Reviewer for this critical remark and the text is extended with new graph (Figure 7 inset) and analysis of significant differences (within the Anisotropic model) between the optical constants (extinction coefficient) calculated at different angles of incidence of light observed at shorter wavelengths. We relate this difference to the notable influence of the small light penetration depth for WTe2 (long/short wavelengths). Using the Beer–Lambert–Bouguer law we have determined the spectral dependence of the penetration depth thus showing that for this particular case the isotropic model gives a better match in determining the extinction coefficient (also valid for the refractive index) at different incidence angles. The penetration depth spectral behavior is visualized by new figure (A3) in the supplementary section and the discussion paragraphs extended accordingly.

Referee 3 Remark 5, 6 and 7

We are very grateful to the Reviewer for pointing the inaccuracy and lack of expression clarity for the Readers in several key paragraphs. Accordingly, we have provided corrected text modifications with more clear physical descriptions.

In the line 248 you write

“The main feature, the observation of negative values for ?′(?) [37] in the ultraviolet/visible edge reveal the surface plasmon polariton activity due to the resonance coupling between the phase of the incident light and the collective excitations of the free carriers.”

but spectral features in UV are due to inter-band electronic transitions (realized by bound electrons), therefore, please, explain in more details what do you mean by … the collective excitation of the free carriers...

Paragraph corrected as follows:

The main feature is the observation of negative values for e'(λ) [37] in the ultraviolet/visible edge revealing the presence of plasmonic resonances mediated by strong inter-band electronic transitions (as alternative mechanism to the collective excitation of the free charge carriers), typically observed in semi-metals (as WTe2), semiconductors, and topological insulator materials [63]

Meaning of the sentence from the line 252

Consequently, the strong optical anisotropy of the dielectric response has a distinct difference in case of single crystals: for a-, b- and c- axis as expected for a Weyl (and Dirac) material with such high uniaxial nature of the Fermi surface and the corresponding directional dielectric polarizability of the carrier inter-band transitions and concentration.

Paragraph corrected as follows:

Consequently, the sophisticated (low symmetry Fermi surface) electronic band structure of Weyl (and Dirac) semi-metal materials results also in the observed notable anisotropy of the optical properties. The corresponding highly directional modulations of the carrier density and mobility leads also to the significant differences in the light polarizability and the dielectric functions along the a-, b- and c- axis in the case of WTe2 single crystals

Meaning of the sentence from the line 160

…the spectroscopic ellipsometry is applied in a transmission regime due to the requirements of Fresnel matrix formalism [55] …

paragraph corrected as follows:

… the spectroscopic ellipsometry is applied in a transmission regime considering the Jones matrix formalism and the finite (non-zero) values of the off-diagonal components …

Thank you for the critical comments and valuable suggestions.

Round 2

Reviewer 3 Report

Please, find my comments in the attached file.

Author Response

Please find the Replay to the Reviewer as attached file since it contain figure and formulas

Round 3

Reviewer 3 Report

Please, find my comments in the attached file.

Author Response

Please find our  response to the reviewer’s comments as attached file. 
